# Ethanol Elevates Excitability of Superior Cervical Ganglion Neurons by Inhibiting Kv7 Channels in a Cell Type-Specific and PI(4,5)P_2_-Dependent Manner

**DOI:** 10.3390/ijms20184419

**Published:** 2019-09-08

**Authors:** Kwon-Woo Kim, Keetae Kim, Hyosang Lee, Byung-Chang Suh

**Affiliations:** 1Department of Brain and cognitive sciences, Daegu Gyeongbuk Institute of Science and Technology (DGIST), Daegu 42988, Korea; 2Department of New biology, Daegu Gyeongbuk Institute of Science and Technology (DGIST), Daegu 42988, Korea

**Keywords:** Kv7 channel, Kv7.2/7.3 current, ethanol, SCG neuron, PI(4,5)P_2_

## Abstract

Alcohol causes diverse acute and chronic symptoms that often lead to critical health problems. Exposure to ethanol alters the activities of sympathetic neurons that control the muscles, eyes, and blood vessels in the brain. Although recent studies have revealed the cellular targets of ethanol, such as ion channels, the molecular mechanism by which alcohol modulates the excitability of sympathetic neurons has not been determined. Here, we demonstrated that ethanol increased the discharge of membrane potentials in sympathetic neurons by inhibiting the M-type or Kv7 channel consisting of the Kv7.2/7.3 subunits, which were involved in determining the membrane potential and excitability of neurons. Three types of sympathetic neurons, classified by their threshold of activation and firing patterns, displayed distinct sensitivities to ethanol, which were negatively correlated with the size of the Kv7 current that differs depending on the type of neuron. Using a heterologous expression system, we further revealed that the inhibitory effects of ethanol on Kv7.2/7.3 currents were facilitated or diminished by adjusting the amount of plasma membrane phosphatidylinositol 4,5-bisphosphate (PI(4,5)P_2_). These results suggested that ethanol and PI(4,5)P_2_ modulated gating of the Kv7 channel in superior cervical ganglion neurons in an antagonistic manner, leading to regulation of the membrane potential and neuronal excitability, as well as the physiological functions mediated by sympathetic neurons.

## 1. Introduction

Alcohol affects diverse physiological functions mediated by the peripheral and central nervous systems [1,2,3] and causes behavioral and psychological symptoms ranging from modest behavioral disinhibition, such as relaxation and motor incoordination [4,5], to severe cognitive dysfunctions, including sedation [6], amnesia [7], hypnosis, and unconsciousness [8]. As high concentrations of ethanol are required to produce physiological changes, early studies proposed that the effects of ethanol occur through perturbation of membrane lipids [9]. However, more recent studies have revealed that alcohol modulates the activity of specific target molecules, including enzymes, ion channels, and receptors [9,10]. For example, ethanol increases the frequency of spontaneous firing in neurons in the ventral tegmental area (VTA) by suppressing Kv7 channel activity [11]. It has also been reported that ethanol activates G protein-gated inwardly rectifying potassium (GIRK) channels by binding to the putative alcohol-binding pocket located near the phospholipid-binding site [12]. In contrast, ethanol inhibits the Shaw2 voltage-gated channel, Kv3, by binding to the amphipathic α-helical region of the intracellular S4-S5 linker [13].

Kv7 channels are voltage-gated potassium channels expressed in many regions throughout the central and peripheral nervous systems [14,15,16]. The Kv7 channel, a heteromer of the Kv7.2 and Kv7.3 subunits, generates slowly activating and non-inactivating currents in response to depolarization of membrane potentials and produces tail currents upon repolarization due to slow deactivation of the channel [16]. As Kv7 channels become activated at membrane potentials higher than −60 mV, they play an important role in determining the resting membrane potential, excitability, and after-hyperpolarization of neurons, as well as theta resonance in the brain [14]. Many studies have demonstrated that plasma membrane phosphatidylinositol 4,5-bisphosphate (PI(4,5)P_2_) is an important cofactor necessary for the gating of Kv7 channels by increasing their open probability [17,18,19]. Thus, hydrolysis of plasma membrane PI(4,5)P_2_ via activation of G_q_-coupled receptors, such as the M_1_ muscarinic acetylcholine receptor (M_1_R), inhibits the channel [20,21]. Potential PI(4,5)P_2_-binding sites have been identified in the S2-S3 and S4-S5 linkers and the proximal region of the C-terminus of Kv7.2/7.3 [22,23].

The superior cervical ganglia (SCG) is a part of the sympathetic nervous system that innervates several regions in the head, including the blood vessels, salivary glands, eyes, choroid plexus, and pineal glands, to control vasculomotor, secretomotor, and pilomotor functions [24,25]. Previous studies have shown that SCG neurons can be classified into three types based on the amplitude of current required to elicit an action potential and their firing patterns [26]. Phasic neurons exhibit discharges of action potentials in response to a current injection mainly during the initial period of the current injection, and they can be further divided into two subclasses. One subset of neurons, the so-called phasic-1 neurons, fires just a few spikes during the initial phase of the stimulatory period regardless of the amplitude of current, whereas the rest or so-called phasic-2 neurons increase their firing frequency in proportion to current intensity. The third type of neuron, the so-called tonic neurons, exhibit high-frequency firing of action potentials throughout the current injection.

Although it is well known that acute and chronic consumption of ethanol can affect the sympathetic system and lead to dysregulation of blood pressure, cardiovascular functions, micturition, and body temperature [27], the effects of ethanol on sympathetic neurons at the molecular and cellular levels remain unclear. In the present study, we demonstrated using patch-clamp recordings that ethanol increased the excitability, as well as resting membrane potential, in SCG neurons in a cell-type selective manner. The effect of ethanol was mediated by inhibiting Kv7 channels and antagonized by increasing the quantity of plasma membrane PI(4,5)P_2_. We also found that Kv7 channels with higher affinity to PI(4,5)P_2_ were only slightly inhibited by ethanol. Thus, our results demonstrated a strong counter-relationship between ethanol and PI(4,5)P_2_ signaling in the modulation of Kv7 channel gating and neuronal excitability.

## 2. Results

### 2.1. SCG Neurons Can Be Classified into Phasic-1, Phasic-2, and Tonic Neurons Based on Their Action Potential Firing Patterns and Kv7 Current (I_Kv7_) Size

To profile SCG neurons electrophysiologically, a series of increasing current pulses from 0 to 180 pA was delivered to primary cultured SCG neurons for 2 s in the current-clamp recordings. Among the 51 SCG neurons recorded, 12 (24%) exhibited phasic-1 neuron-like responses in which just a few spikes were elicited immediately after applying the current, and the number of spikes did not substantially increase when a high amplitude of current, such as 180 pA, was delivered (Figure 1A–C; phasic-1 or P1). Another 21 neurons (41%) displayed phasic-2 neuron-like responses in which a small number of spikes was evoked by a weak amplitude of the current (Figure 1A–C; phasic-2 or P2). The firing frequency of this type of neuron increased gradually in proportion to the amplitude of the injected current. The remaining 18 neurons (35%) displayed tonic neuron-like responses in which a low amplitude of current, such as 60 pA, was sufficient to evoke a high frequency of firings throughout the current injection (Figure 1A–C; tonic or T). Tonic neurons reached their maximum firing frequency when 140 pA was injected. A comparison of the resting membrane potentials revealed that the three types of SCG neurons were maintained at different potentials, in which phasic-1 and tonic neurons exhibited the lowest and highest resting membrane potentials, respectively (Figure 1D). Besides, tonic neurons had a significantly lower activation threshold than phasic neurons as demonstrated by the rheobase, the lowest amplitude of an injected current required to evoke an action potential (Figure 1E). No difference was observed in membrane conductance (C-slow) among the three types of SCG neurons, indicating that the sizes of the SCG neurons were similar (Figure 1F).

We also found an inverse correlation between the firing frequency and the size of the Kv7 current (*I*_Kv7_) in different types of SCG neurons (Figure 1G). Tonic neurons that fired at high frequency in response to an injection of 100 pA exhibited *I*_Kv7_ smaller than 10 pA/pF, while phasic-1 and phasic-2 neurons, which discharge at low frequencies, showed variable *I*_Kv7_ ranging from 3 to 30 pA/pF. Taken together, these results were consistent with previous findings that SCG neurons could be broadly classified into three different types according to their electrophysiological properties, including the firing patterns of action potentials, the resting membrane potential, and activation threshold, which is closely related to *I*_Kv7_.

### 2.2. Excitability and the Resting Membrane Potential of SCG Neurons Are Affected by Inhibiting Kv7 Channels in a Cell Type-Specific Manner

To further determine the role of Kv7 channels in regulating the excitability of SCG neurons, Kv7 channels were inhibited either by hydrolyzing plasma membrane PI(4,5)P_2_ through M_1_R activation using the agonist oxotremorine-M (Oxo-M, 10 μM) or by treating with XE991 (10 μM), a selective inhibitor of Kv7 channels (Figure 2). Such treatments significantly increased the number of spikes evoked by an injection of 60 pA in all three types of SCG neurons. Oxo-M was more potent at increasing the number of action potentials in phasic-1 and phasic-2 neurons than tonic neurons, indicating that Kv7 channels were more abundant in phasic cells than tonic cells (Figure 2A–C). The resting membrane potentials of phasic-1 and phasic-2 neurons increased significantly in response to the Oxo-M or XE991 treatment (Figure 2D). Such an increase was not observed in tonic neurons (Figure 2D). Taken together, these results indicated that Kv7 channels modulated the resting membrane potential, as well as the excitability of SCG neurons. In correlation with the size of *I*_Kv7_, shown in Figure 1G, Kv7 channel inhibitors caused more dramatic increases in phasic neurons than in tonic neurons.

### 2.3. Ethanol Increases the Excitability of SCG Neurons

We then examined whether ethanol modulates the excitability of SCG neurons by measuring action potentials elicited by an injection of 60 pA while the cells were incubated with 3, 10, 30, or 100 mM ethanol. The pharmacologically relevant concentration is up to 100 mM, which is converted to the blood alcohol concentration of ~0.5% for the high possibility of death [28]. The ethanol treatment did not cause the spontaneous discharge of SCG neurons in the absence of current injection. However, SCG neurons exhibited the firing of action potentials that were potentiated by exposure to ethanol in response to a 60 pA injection (Figure 3A). Compared with untreated controls, phasic-1 and phasic-2 neurons showed slight and moderate increases in the number of discharges in the presence of a high concentration of ethanol. In contrast, tonic neurons exhibited a relatively larger increase in firings than the other types (Figure 3A–C, bottom). This result indicated that ethanol augmented the excitability of SCG neurons and that the effect of ethanol was dependent on the type of SCG neuron. The number of action potentials increased by ethanol was smaller than that induced by Oxo-M, indicating that the modulatory effect of ethanol was much weaker than Oxo-M (Figure 2B,C vs. Figure 3B,C).

To investigate the molecular mechanism underlying the increased excitability of SCG neurons in response to ethanol, we compared the medium after-hyperpolarization (mAHP) elicited by an injection of 60 pA because *I*_Kv7_ plays a key role in determining excitability of the SCG neurons. mAHP is a sudden drop in the membrane potential at the cessation of the current injection, mainly contributed by slowly deactivating Kv7 channels [14,29]. Compared with untreated controls, exposure to ethanol led to a significant reduction in mAHP in tonic neurons at concentrations of 30 and 100 mM, and also caused a small but significant decrease in mAHP in phasic-2 neurons in response to 100 mM ethanol. mAHP was unchanged by ethanol in phasic-1 neurons (Figure 3D–F). Thus, the inhibitory effect of ethanol was negatively correlated with the size of *I*_Kv7_, and, therefore, tonic neurons with a smaller *I*_Kv7_ value might be more strongly inhibited by a weak modulator, ethanol, than phasic neurons with a larger *I*_Kv7_. Taken together, these results suggested that ethanol increased the firing of SCG neurons in a cell type-specific manner by inhibiting Kv7 channels, which play a critical role in determining the resting membrane potential and excitability of neurons.

### 2.4. Ethanol Inhibits Kv7 Channels in Cultured SCG Neurons as Well as the Heterologous Kv7.2/7.3 Channels Expressed in TSA201 Cells

To further examine whether ethanol increases the firing frequency of SCG neurons by inhibiting Kv7 channels, primary cultured SCG neurons were applied with −20 mV depolarizing pulse in whole-cell patch-clamp recordings to elicit *I*_Kv7_ in SCG neurons. Kv7 currents were calculated by subtracting the amplitude of the current measured at 10 ms before the end of the tail current from that measured at 5–10 ms after the peak of the current, which is presented as *I*_Kv7_(Y_0_ − Y_1_) (Figure 4A). The time-course analysis demonstrated that *I*_Kv7_(Y_0_ − Y_1_) was diminished by 18 ± 2% and 27 ± 3% in response to 100 and 200 mM ethanol, respectively, and recovered partially to the pretreatment level after washing out of the ethanol (Figure 4B,C). Oxo-M (10 µM) was more potent in decreasing *I*_Kv7_ (Y_0_ − Y_1_) of SCG neurons than ethanol (Figure 4B,C). To further determine whether ethanol inhibits Kv7 currents, human embryonic kidney-derived tsA201 cells were transiently transfected with a mixture of cDNAs encoding Kv7.2 and Kv7.3. The Kv7.2/7.3 currents (*I*_Kv7.2/7.__3_) were elicited by applying −20 mV to tsA201 cells. Consistent with the results obtained from SCG neurons, the time-course analysis showed that *I*_Kv7.2/7.__3_ was inhibited by 16 ± 2% and 52 ± 5% in the presence of 100 and 400 mM ethanol, respectively (Figure 4D,E). The inhibitory effect of ethanol was strengthened in a dose-dependent manner (Figure 4F). When 400 mM ethanol was used, the current-voltage relationship of cells expressing Kv7.2/7.3 shifted downwards, but not in the left or right directions, indicating that the voltage sensitivity of Kv7.2/7.3 was unaffected by ethanol (Figure 4G). Taken together, these results indicated that ethanol inhibited *I*_Kv7_ of SCG neurons, as well as that of cells transfected with Kv7.2/7.3, and suggested that the increased discharge of action potentials in SCG neurons in response to ethanol was mediated by inhibiting the Kv7 channel.

### 2.5. The Inhibition of I_Kv7_ by Ethanol Is Not Due to Either Degradation or Translocation of Plasma Membrane PI(4,5)P_2_

As plasma membrane PI(4,5)P_2_ is one of the main activators of Kv7.2/7.3, we next examined whether ethanol inhibits *I*_Kv7.2/7.__3_ by decreasing the amount of plasma membrane PI(4,5)P_2_. The PI(4,5)P_2_-binding fluorescent protein, the pleckstrin homology domain of phospholipase C-δ1 (PH(PLCδ1)-GFP), was expressed in tsA201 cells together with M_1_R, and the localization of PH(PLCδ1)-GFP was continuously monitored using confocal microscopy while the cells were treated with Oxo-M or ethanol (Figure 4H). GFP fluorescence was predominantly observed in the plasma membrane before the drug treatment. Consistent with previous reports, PH(PLCδ1)-GFP was translocated into the cytosolic compartment in response to 10 µM Oxo-M, which activated phospholipase C upon binding to M_1_R (Figure 4H,I). In contrast, 400 mM ethanol caused no detectable changes in the localization of PH(PLCδ1)-GFP, and co-application with Oxo-M failed to further increase fluorescence intensity in the cytosolic compartment compared with Oxo-M only (Figure 4H,I). These results suggested that ethanol did not cause either a detectable decrease in the amount of plasma membrane PI(4,5)P_2_ or its translocation into the cytosolic compartment, and suggested that the inhibitory effect of ethanol on *I*_Kv7.2/7.__3_ was not due to a reduction in the level of PI(4,5)P_2_ in the plasma membrane.

### 2.6. Inhibitory Effect of Ethanol on I_Kv7.2/7.__3_ is Modulated by the Level of the Plasma Membrane PI(4,5)P_2_

To test whether the inhibitory effect of ethanol on *I*_Kv7.2/7.__3_ was affected by the amount of plasma membrane PI(4,5)P_2_, phosphatidylinositol phosphate 5-kinase Iγ (PIPKIγ), the enzyme that increases the level of PI(4,5)P_2_ by phosphorylating phosphatidylinositol 4-phosphate (PI(4)P) [30], was overexpressed in tsA201 cells together with Kv7.2/7.3 and M_1_R. Consistent with our previous findings, overexpressing PIPKIγ led to a leftward shift in the I-V relationship of tail currents by 10 mV and a significant increase in the deactivation rate, as well as augmentation of *I*_Kv7.2/7.__3_ in response to −60 mV, compared to control cells absent of PIPKIγ ([31], Figure 5A,B). This result indicated that PIPKIγ was functionally expressed in transfected cells. The time-course analysis showed that *I*_Kv7.2/7.__3_ was inhibited by 400 mM ethanol, as well as the broad spectrum K^+^ channel blocker tetraethylammonium (TEA, 30 mM) and 10 µM Oxo-M. The inhibitory effect of those drugs was strongly diminished by overexpressing PIPKIγ, in that ethanol was most strongly affected (Figure 5C,D). These results suggested that the inhibitory effect of ethanol on *I*_Kv7.2/7.__3_ was antagonized by a high level of plasma membrane PI(4,5)P_2_.

It has been shown that the Kv7 subtypes display variable sensitivities to PI(4,5)P_2_ [18,32]. For example, Kv7.2 is weak, Kv7.2/7.3 is moderate, and Kv7.3 is strong in relative affinity to PI(4,5)P_2_. By taking advantage of this characteristic, we examined whether there is an inter-relationship between the affinity of Kv7 subtypes to PI(4,5)P_2_ and their sensitivity to ethanol. TsA201 cells expressing Kv7.2 and Kv7.3 either individually or in combination were electrophysiologically stimulated to elicit *I*_Kv7.2/7.3_. In response to 200 and 400 mM ethanol for 40 s, cells expressing homomeric and heteromeric channels displayed reduced current to a variable degree, which was inversely correlated with their reported affinity to PI(4,5)P_2_ (Figure 6A,B). Heteromeric channels had intermediate sensitivity to inhibition by ethanol. An increase in the amount of plasma membrane PI(4,5)P_2_ by overexpressing PIPKIγ mitigated the inhibitory effect of ethanol, but the Kv7.2 channels were more sensitive to the inhibition (Figure 6D). 2-Propanol also inhibited Kv7-mediated currents, and the extent of inhibition was inversely correlated with the affinity of those channels to PI(4,5)P_2_ (Figure 6E).

As an alternative approach to assessing the relationship between plasma membrane PI(4,5)P_2_ and the inhibitory effect of ethanol on *I*_Kv7.2/7.3_, plasma membrane PI(4,5)P_2_ was acutely hydrolyzed by activating the voltage-sensing phosphatase, *Ciona intestinalis* VSP (Ci-VSP), which cleaves the 5-phosphate from the inositol ring of PI(4,5)P_2_ in response to depolarization of the membrane potential [33]. TsA201 cells expressing both Ci-VSP and a mixture of Kv7.2 and Kv7.3 were depolarized to +30 mV for 42 s, which caused a gradual reduction in *I*_Kv7.2/7.3_ (Figure 7A). *I*_Kv7.2/7.3_ was further diminished by treatment with 400 mM ethanol, as measured by τ, a time constant (Figure 7B). In another experiment using a different type of voltage-sensing phosphatase, *Danio rerio* VSP (Dr-VSP), we confirmed facilitation of τ upon treatment with 400 mM ethanol and observed a delay in the recovery of the *I*_Kv7.2/7.3_ following inactivation of Dr-VSP achieved by returning the membrane potential to −20 mV (Figure 7C–F). Taken together, these results suggested that there was a negative inter-relationship between plasma membrane PI(4,5)P_2_ and ethanol in the modulation of *I*_Kv7.2/7.3_, which might be the molecular basis responsible for the increased excitability of SCG neurons elicited by ethanol.

## 3. Discussion

In this study, we demonstrated that ethanol enhanced excitability and inhibited the mAHP in SCG neurons by inhibiting the Kv7 current. Among the three types of SCG neurons identified in this study, ethanol modulated the discharge of action potentials most strongly in the tonic type of SCG neurons, which expressed a relatively smaller amount of the Kv7 channel. We further demonstrated that the amount of plasma membrane PI(4,5)P_2_ and the affinity between Kv7 channels and PI(4,5)P_2_ modulated the inhibitory effect of ethanol on the Kv7 channel. This is the first study to use SCG neurons and show the cell type-specific modulatory effect of ethanol and the interactive mode of regulation between ethanol and phospholipid in the gating of Kv7 channels.

Ethanol has been shown to regulate neuronal excitability and synaptic transmission through activation or inhibition of ligand- and voltage-gated ion channels. Upon treatment with ethanol, γ-Aminobutyric acid (GABA)-mediated inhibitory synaptic transmission is potentiated in the hippocampus, as well as the cerebellum, while *N*-methyl-d-aspartate (NMDA)-mediated excitatory transmission is diminished in the hippocampus [34,35]. It has been reported that ethanol inhibits glycine receptors to potentiate tonic inhibitory synaptic transmission in the orbitofrontal cortex, whereas it augments glycine-mediated currents and facilitates depolarization of neurons in the ventral tegmental area (VTA) [36,37]. Also, ethanol promotes the gating of hyperpolarization-activated cation channels (HCNs) to increase the spontaneous firing of dopaminergic neurons in the hippocampus and VTA [38,39,40]. It has also been demonstrated that ethanol increases the open probability of the large-conductance, calcium-activated-K^+^ (BK) channel to reduce action potential discharges in medium spiny neurons of the striatum as well as the low-frequency pacemaker neurons in the external globus pallidus [41,42]. The Kv7 channel has also been demonstrated to be regulated by ethanol. Ethanol inhibits Kv7 currents in hippocampal CA1 neurons, as well as dopaminergic neurons in the VTA, leading to an increase in the resting membrane potential and excitability of those neurons [11,43]. Consistent with these observations, we found here that ethanol inhibited the Kv7 current, increasing the resting membrane potential and excitability of SCG neurons. We also revealed that the susceptibility to ethanol varied depending on the type of SCG neuron, in that a low concentration of ethanol (30 mM) selectively promoted the firing of action potentials in tonic SCG neurons, whereas a high concentration of ethanol (100 mM) elicited an increase in discharge of action potentials from both tonic and phasic neurons. As ethanol is a weak modulator of Kv7 channels that inhibits the Kv7 current of SCG neurons by 10% at a concentration of 30 mM, and tonic neurons express a lower level of M-channels than phasic neurons, such partial inhibition could be more effective in driving changes in tonic neurons in which a small number of Kv7 channels play a critical role in determining the resting membrane potential and neuronal excitability [26]. In addition to the M-channel, the effect of ethanol on the excitability of SCG neurons may involve other ion channels that are sensitive to ethanol, such as BK, small-conductance, calcium-activated-K^+^ (SK), and HCN channels [44,45,46,47,48]. A previous study performed on dissociated SCG neurons showed that ethanol inhibits, rather than promotes, the firing of action potentials in those neurons by suppressing Na^+^ current [49]. Although we observed a similar reduction in firing from a small fraction of SCG neurons (five of 56 cells), the vast majority displayed an increased discharge rate in response to ethanol. As that study did not provide detailed information on the experimental conditions, including the fraction and type of SCG neuron showing such a reduction, it is difficult to determine the reasons for the discrepancy between the studies.

What is the structural mechanism by which ethanol regulates the gating of Kv7 channels? A high-resolution structural study on Kir2.1 and inwardly rectifier K^+^ channel 1 (IRK1), the *Drosophila* GIRK homolog, has identified three hydrophobic alcohol-binding pockets formed by the N-terminus and the βD-βE and βL-βM loops of the C-terminus [50]. Accordingly, a leucine mutation within the putative GIRK2 alcohol-binding pocket deduced from that of IRK1 to a bulkier tryptophan diminishes alcohol-elicited potentiation of the GIRK2 current, confirming the validity of the identified alcohol-binding pocket [51,52]. A recent structural study on alcohol-sensitive Shaw2 and its alcohol-insensitive human homolog Kv3.4 demonstrated that an intracellular domain formed by the S4-S5 linker and S6 plays a critical role in determining their alcohol sensitivity [13,53,54]. Substitution of glycine 371 in the S4-S5 linker of Kv3.4 to a comparable amino acid in Shaw2, isoleucine, is sufficient to switch Kv3.4 to a channel that is sensitive to alcohol [53]. However, a mutation in proline 410 in S6 of Shaw2 to alanine changes the outcome of alcohol modulation from inhibition to potentiation [54]. Given the structural similarity between Shaw2 and the Kv7 channel, sequence comparison of the two ion channels and the site-directed mutagenesis of residues within the putative alcohol-binding pockets in the Kv7 channel may facilitate understanding the structural mechanism underlying the regulation of Kv7 channel gating by ethanol.

Another interesting finding we made in this study was the antagonistic functional interaction between ethanol and plasma membrane PI(4,5)P_2_ in regulating gating of the Kv7 channel. It has been shown that the Kv7 current is inhibited by ethanol, but promoted by plasma membrane PI(4,5)P_2_ [17,19,20,23,55]. Here, we found that the inhibitory effect of ethanol on the Kv7 current was altered when either the amount of PI(4,5)P_2_ or the affinity of the Kv7 channel to the phospholipid increased. Furthermore, ethanol facilitated a reduction in the Kv7 current when combined with the hydrolysis of plasma membrane PI(4,5)P_2_. In contrast, both modulators promote the gating of GIRK channels so that depletion of plasma membrane PI(4,5)P_2_ attenuates the GIRK current induced by ethanol, while ethanol slows down the reduction in the GIRK current caused by dephosphorylation of plasma membrane PI(4,5)P_2_ [12]. PI(4,5)P_2_ has been reported to bind to the tether helix of the C-linker in the GIRK2 channel, which is located near the N-terminus and the second transmembrane domain [56]. It has been suggested that binding of ethanol to the hydrophobic pocket of the GIRK channel strengthens affinity of the GIRK channel to PI(4,5)P_2_, stabilizing the open state of the ion channel [57]. A previous study identified that PI(4,5)P_2_ binds to the cytosolic domain formed by the S2-S3 and S4-S5 linkers, as well as the proximal C-terminus of the Kv7 channel, using charge-neutralizing or reversing mutations on basic residues in the Kv7 channel [17,55]. Mutations in the amino acids that are critical for PI(4,5)P_2_ binding reduce the affinity of the mutant channels to PI(4,5)P_2_ but leave their voltage sensitivity unaffected [23]. As the amino acid residues and the domains responsible for the modulatory effect of ethanol have yet to be identified in the Kv7 channel, it is unclear how ethanol and PI(4,5)P_2_ interact and lead to opposite results in gating of the ion channel. One possibility is that ethanol and PI(4,5)P_2_ bind to separate domains in the Kv7 channel and act independently to regulate gating of the channel. Another possibility is that there is a structural interaction between alcohol- and the PI(4,5)P_2_-binding domains and that binding of ethanol may interfere with the interaction between the ion channel and PI(4,5)P_2_, or vice versa. Alternatively, two modulators may compete for an overlapping domain to bind to the Kv7 channel. The first step to reveal the structural mechanism for the interaction between ethanol and PI(4,5)P_2_ is to identify the amino acid residues and the domains responsible for the modulatory effect of ethanol.

Acute exposure to ethanol affects the physiological functions mediated by SCG neurons, including secretion from salivary glands, vasoconstriction, and pilomotor reactions [25]. Chronic exposure to ethanol reduces viability, causes morphological changes, and elicits extensive vacuolation of SCG neurons [2,58]. Therefore, determining the regulatory mechanism of suppressed Kv7 channel function in response to ethanol will provide an important clue to understanding such acute and chronic effects of ethanol in SCG neurons.

## 4. Materials and Methods

### 4.1. Cell Culture and Transfection

TsA201 cells, a transformed human kidney 293 cell line stably expressing an SV40 temperature-sensitive T antigen (from Bertil Hille, University of Washington School of Medicine, Seattle, WA, USA), were cultured in Dulbecco’s Modified Eagle Medium (Invitrogen, Carlsbad, CA, USA) with 0.2% penicillin/streptomycin (HyClone, Thermo Fisher Scientific) and 10% fetal bovine serum (Invitrogen, Carlsbad, CA, USA). The cells were subcultured at a density of 2% every 4–5 days using Dulbecco’s Phosphate-Buffered Saline (DPBS: Gibco, Thermo Fisher Scientific) to detach the cells. Kv7.2 (GenBank accession number AF110020) and Kv7.3 (accession number NM_004519) channels were transiently transfected at a 1:1 ratio with 0.2 μg Ds-Red as a transfection marker to measure the Kv7.2/7.3 current. In some experiments, 0.8–1 μg M_1_R (accession number NM_080773), 0.5–1 μg Dr-VSP with internal ribosome entry site (IRES)-EFGP, or 0.5–1 μg Ci-VSP (both VSPs from Yasushi Okamura, Osaka University, Japan) were cotransfected. PIP 5-kinase type Iγ (0.5 μg, PIPKIγ; provided by Y. Aikawa and T. F. Martin, University of Wisconsin, Madison, WI, USA) was cotransfected to increase the plasma membrane PI(4,5)P_2_ level. For the confocal experiments, 0.3 μg PH(PLCδ1)-GFP (from Bertil Hille, University of Washington School of Medicine, Seattle, WA, USA) was cotransfected to label membrane PI(4,5)P_2_. Lipofectamine 2000 (Invitrogen, Carlsbad, CA, USA) was used to transfect cDNA when confluency of the cells reached 50–70% in a 35-mm dish. The transfected tsA201 cells were plated onto coverslip chips coated with 0.1 mg/mL poly-L-lysine (Sigma-Aldrich, St. Louis, MO, USA), and the fluorescent cells were studied during electrophysiological and confocal experiments 36–48 h after transfection, as described previously [31].

Animal experiments were approved by the Institutional Animal Care and Use Committee at Daegu Gyeongbuk Institute of Science and Technology (Approval Number: DGIST-IACUC-19060304-01, 3 June 2019). SCG neurons were cultured from 5–7 days postnatal Sprague–Dawley using a protocol described previously [59]. In brief, the rats were anesthetized with CO_2_ and quickly decapitated. Ganglia were incubated with 0.25% trypsin solution at 37 °C for 20 min. Following the enzyme treatment, the ganglia were suspended in RPMI 1640 medium plus 10% heat-inactivated horse serum and 5% fetal bovine serum to stop digestion. They were resuspended in RPMI 1640 medium plus 1% heat-inactivated horse serum, nerve growth factor, penicillin/streptomycin, and uridine/5-fluorodeoxyuridine. The cells were plated onto 0.1 mg/mL poly-L-lysine chips, incubated at 37 °C (5% CO_2_) overnight, and used within 3 days.

### 4.2. Electrophysiological Recording

TsA201 cells were whole-cell clamped at room temperature (22–25 °C). The electrodes were pulled from borosilicate glass micropipette capillaries (1.5 mm in outside diameter (OD); 1.10 mm in internal diameter (ID); 10 cm in length; Sutter Instrument, Novato, CA, USA) using a P-97 micropipette puller (Sutter Instrument, Novato, CA, USA) with resistances of 1.8–4 MΩ. Recordings were performed using a HEKA EPC-10 amplifier and Pulse software (HEKA Elektronik, Pfalz, Germany). The whole-cell access resistance was 2–6 MΩ, and series-resistance errors were compensated by 60%. Recording started 3–5 min after breakthrough. The Kv7 current was studied by holding the cell at –20 mV and applying a 500 ms hyperpolarizing step to –60 mV every 2 or 4 s. The data analysis was done using Igor Pro 6.0 (WaveMetrics, Inc., Tigard, OR, USA). The τ values of deactivating tail current and activating current were obtained by fitting each trace with a single exponential function at –60 mV and –20 mV, respectively. The holding potential was −80 mV in all voltage protocols, except where indicated.

Kv7 currents in SCG neurons were measured in a whole-cell configuration at room temperature with 4–7 MΩ pipette resistance. The current was measured from deactivation current records at −60 mV as the difference between the average of a 10 ms segment, taken 10–20 ms into the hyperpolarizing step, and the average during the last 10 ms of that step. Action potentials of SCG neurons were recorded using current-clamp mode. The current injection was held at 0 and elicited for 2 s.

### 4.3. Confocal Imaging and Analysis

Confocal microscopic images were taken every 5 s on the Carl Zeiss inverted LSM 700 confocal microscope (Carl Zeiss AG, Jena, Germany) at room temperature. The experiments were performed using Ringer’s solution. Cytosolic fluorescence intensities were analyzed with the ROI manager tool in ImageJ (National Institutes of Health, Bethesda, MD, USA) and IgorPro (WaveMetrics, Inc., Tigard, OR, USA) and with Excel 2016 (Microsoft Inc., Redmond, WA, USA). To obtain the average time courses, the fluorescence intensity *F* over a given region of the cytoplasm was normalized to the average baseline intensity for 30 s before applying the agonist *F*_0_ (*F*/*F*_0_).

### 4.4. Solutions and Materials

The Ringer’s solution was used to record the K^+^ current and for confocal microscopy. It consisted of 160 mM NaCl, 2.5 mM KCl, 2 mM CaCl_2_, 1 mM MgCl_2_, 10 mM HEPES, and 8 mM glucose, adjusted to pH 7.4 with NaOH. The pipette solution contained 175 mM KCl, 5 mM MgCl_2_, 5 mM HEPES, 0.1 mM BAPTA, 3 mM Na_2_ATP, and 0.1 mM Na_3_GTP, then adjusted to pH 7.4 with KOH.

### 4.5. Data Analysis

Data were analyzed with Excel 2016 and IgorPro. The time constants were measured by exponential fit. All quantitative data are expressed as mean ± standard error. One-way analysis of variance (ANOVA) followed by Sidak’s posthoc test and Mann–Whitney *u*-test and Wilcoxon matched-pairs signed-rank were performed. A *p*-value <0.05 was considered significant.

## Figures and Tables

**Figure 1 ijms-20-04419-f001:**
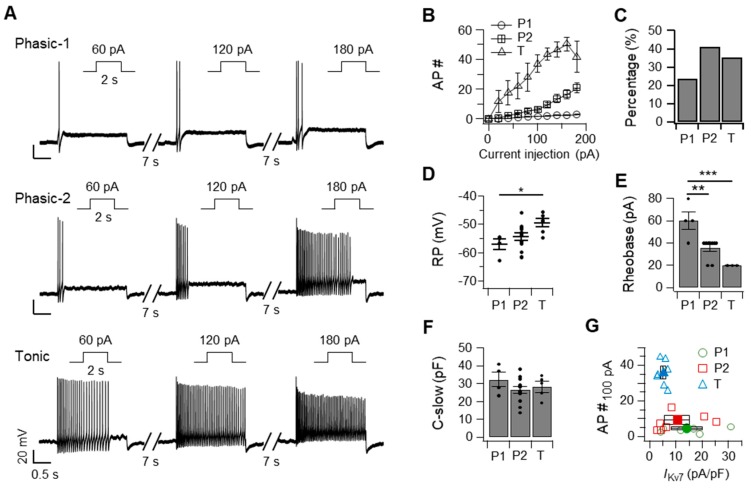
Superior cervical ganglia (SCG) neurons can be classified as phasic-1, phasic-2, and tonic cells, based on their firing pattern, resting membrane potential, and Kv7 current density. Representative traces showing discharges (**A**) and the average number (**B**) of action potentials elicited by an injection of the indicated current for 2 s in phasic-1, phasic-2, and tonic neurons. (**C**) Percentage of recorded neurons classified as the indicated cell types. The resting membrane potential (RP, **D**), rheobase (**E**), and C-slow (**F**) of each cell type. (**G**) Relationship between the density of Kv7 currents and the number of action potentials triggered by an injection of 100 pA into phasic-1 (circles), phasic-2 (squares), and tonic (triangles) neurons (P1, *n* = 6; P2, *n* = 7; T, *n* = 7). Black symbols indicate the average. Data are mean ± SEM. One-way analysis of variance followed by Sidak’s posthoc test, * *p* < 0.05, ** *p* < 0.01, *** *p* < 0.001.

**Figure 2 ijms-20-04419-f002:**
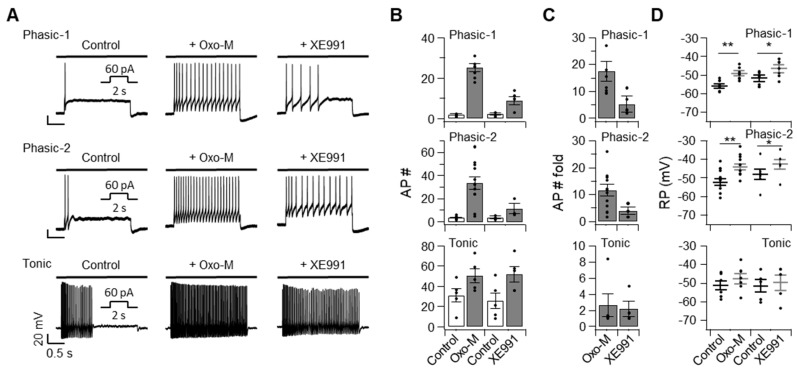
Three types of superior cervical ganglia (SCG) neurons exhibiting differential firing responses and membrane potentials upon inhibition of the Kv7 current. (**A**) Representative traces, showing action potential discharges elicited by an injection of 60 pA for 2 s into each cell type incubated with 10 μM oxotremorine-M (Oxo-M) or 30 μM XE991, compared with the untreated control. The number (**B**), the fold-change of the action potentials (**C**), and the resting membrane potential (**D**) shown by each cell type in the absence (control) or presence of Oxo-M or XE991 (*n* = 3–11). Data are mean ± SEM. Wilcoxon matched-pairs signed-rank test. * *p* < 0.05, ** *p* < 0.01.

**Figure 3 ijms-20-04419-f003:**
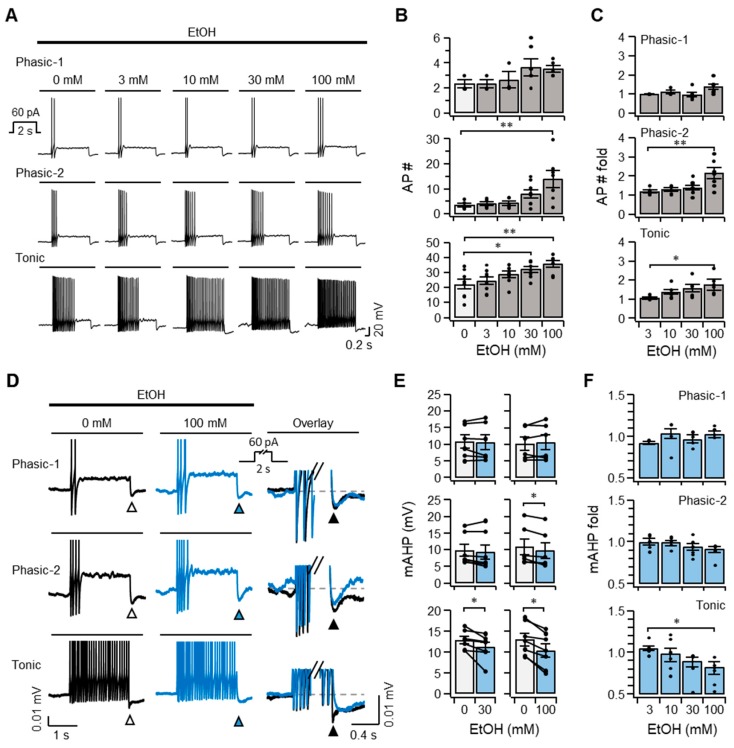
Superior cervical ganglia (SCG) neurons exhibit an increase in action potential discharges and a decrease in medium after-hyperpolarization (mAHP) in response to ethanol in a cell type-specific manner. (**A**) Representative traces, showing action potential discharges elicited by an injection of 60 pA for 2 s into each cell type incubated with the indicated concentrations of ethanol for 40 s. The number (**B**) and fold-change of the action potentials (**C**) shown by each cell type in the absence or presence of ethanol (*n* = 3–8). (**D**) Representative traces showing mAHP (triangles) elicited by an injection of 60 pA for 2 s into each cell type in the absence or presence of 100 mM ethanol for 40 s. The right trace shows the composite mAHP with or without ethanol. mAHP of individual samples (**E**) and the fold-change of mAHP (**F**) shown by each cell type in the absence (black symbols) or presence of ethanol (blue symbols). *n* = 3–7. Data are mean ± SEM. One-way analysis of variance (ANOVA) followed by Sidak’s posthoc test (**B**,**C**, and **F**) and Wilcoxon matched-pairs signed-rank test (**E**). * *p* < 0.05, ** *p* < 0.01.

**Figure 4 ijms-20-04419-f004:**
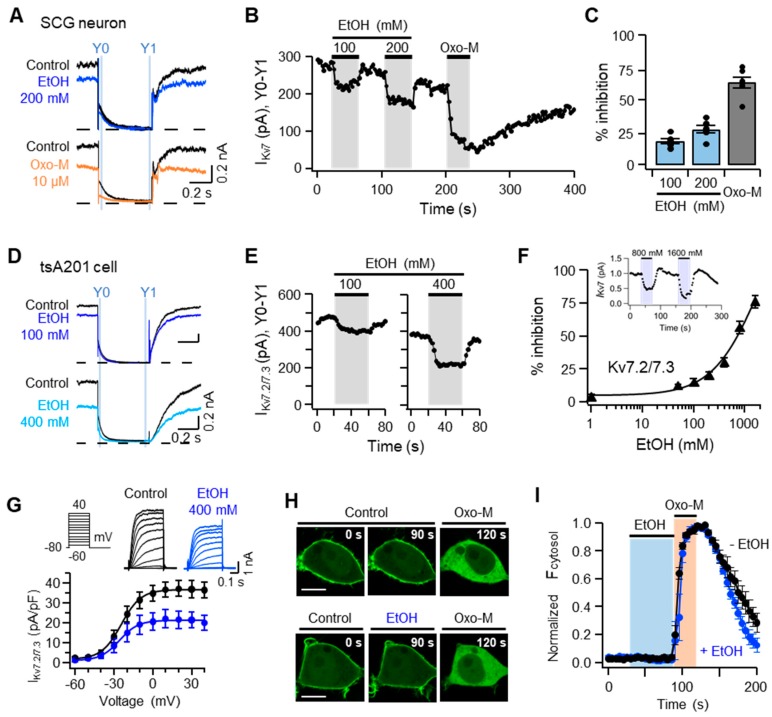
Ethanol inhibits Kv7 currents in superior cervical ganglia (SCG) neurons as well as in tsA201 cells transfected with Kv7.2/7.3. (**A**) Representative traces, showing Kv7 currents elicited in primary cultured SCG neurons using the protocol of −20 to −50 mV, while the cells were incubated with either 200 mM ethanol or 10 µM Oxo-M. The dashed line indicates the zero-current level. (**B**) Time-course analysis of *I*_Kv7_(Y_0_ − Y_1_) upon sequential treatments of 100 and 200 mM ethanol for 40 s each, followed by 10 µM Oxo-M for 30 s. (**C**) Inhibition of Kv7 currents in primary cultured SCG neurons after treatment with ethanol or Oxo-M (*n* = 6). (**D**) Representative traces, showing Kv7.2/7.3 currents elicited in tsA201 cells in the presence of ethanol (100 and 200 mM) or 10 µM Oxo-M. (**E**) Time-course of Kv7.2/7.3 currents upon sequential treatments of 100 and 200 mM ethanol for 40 s each. (**F**) Dose-dependency, showing the inhibitory effect of ethanol on the Kv7.2/7.3 current in tsA201 cells (*n* = 5–9 for each dose). Cells were treated with 1, 50, 100, 200, 400, 800, or 1600 mM ethanol, and the inhibition of Kv7 current was measured. Inset, Time-course of Kv7 current regulation by 800 and 1600 mM ethanol. (**G**) Current-voltage relationship of Kv7.2/7.3 currents elicited in tsA201 cells using increasing voltage steps from −60 mV to +40 mV while the cells were untreated (black symbols, *n* = 4) or treated with 400 mM ethanol (blue symbols, *n* = 4). (**H**) Live confocal imaging showing intracellular localization of native fluorescence of pleckstrin homology domain of phospholipase C-δ1 labeled with green fluorescence protein (PH(PLCδ1)-GFP) within the tsA201 cells before and after treatment with 400 mM ethanol for 60 s, followed by 10 µM Oxo-M for 30 s. Scale bars, 10 µm. (**I**) Native fluorescence of PH(PLCδ1)-GFP in the cytosolic compartment normalized by the maximum value of fluorescence obtained during Oxo-M treatment and the minimum in control (*n* = 5 each).

**Figure 5 ijms-20-04419-f005:**
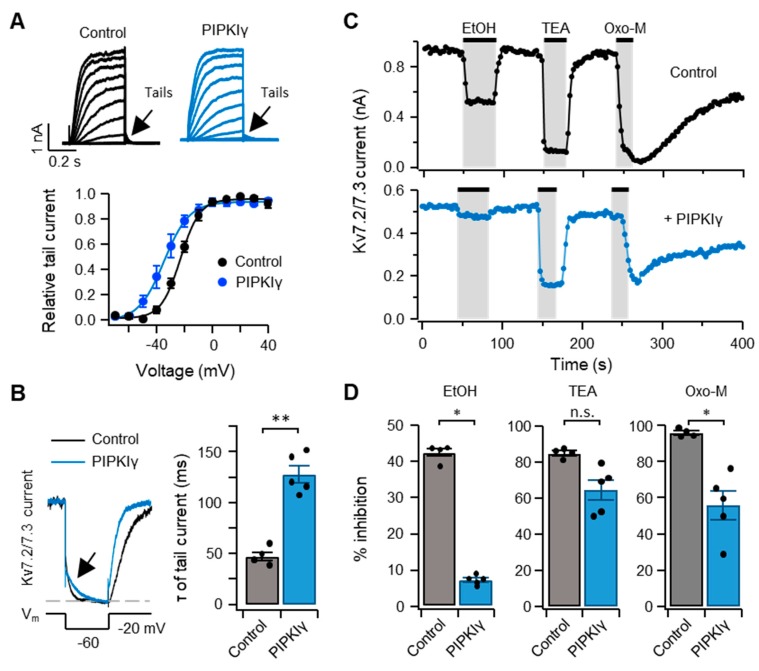
Kv7.2/7.3 currents in transfected tsA201 cells are inhibited by an increase in plasma membrane phosphatidylinositol 4,5-bisphosphate (PI(4,5)P_2_) using phosphatidylinositol phosphate 5-kinase type 1γ (PIPKIγ). (**A**) Tail currents elicited using increasing voltage steps applied to tsA201 cells untransfected (black symbols) or overexpressing PIPKIγ (blue symbols). The holding potential was −80 mV. (**B**) Representative traces and the time constant for deactivating of Kv7.2/7.3 currents exhibited by tsA201 cells untransfected (black symbols) or overexpressing PIPKIγ (blue symbols). Time-course (**C**) and percent inhibition (**D**) of Kv7.2/7.3 currents upon sequential treatments of 400 mM ethanol, 30 mM tetraethylammonium (TEA), and 10 μM Oxo-M in tsA201 cells untransfected (control) or overexpressing PIPKIγ. Data are mean ± SEM. *n* = 5 each. Mann–Whitney *u*-test, * *p* < 0.05, ** *p* < 0.01.

**Figure 6 ijms-20-04419-f006:**
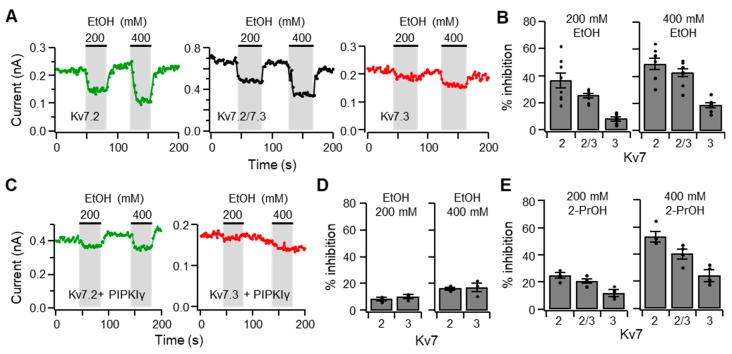
The inhibitory effect of ethanol is negatively correlated with the affinity of Kv7 subtypes to PI(4,5)P_2_. (**A**) Representative Kv7 currents exhibited by tsA201 cells transfected with Kv7.2 and Kv7.3 either individually or in combination. The currents were measured every 2 s in the presence of 200 and 400 mM ethanol for 40 s. (**B**) Percent inhibition of Kv7 currents in tsA201 cells incubated with ethanol (*n* = 8 each). (**C**) Representative traces of tsA201 cells transfected with either Kv7.2 (green) or Kv7.3 (red) in combination with PIPKIγ. (**D**) Summary of Kv7 current inhibition in tsA201 transfected with either Kv7.2 or Kv7.3 in combination with PIPKIγ (*n* = 3 each). (**E**) Percent inhibition of Kv7 currents in tsA201 cells upon treatment with 200 or 400 mM 2-propanol (2-PrOH, *n* = 5 each). Data are mean ± SEM.

**Figure 7 ijms-20-04419-f007:**
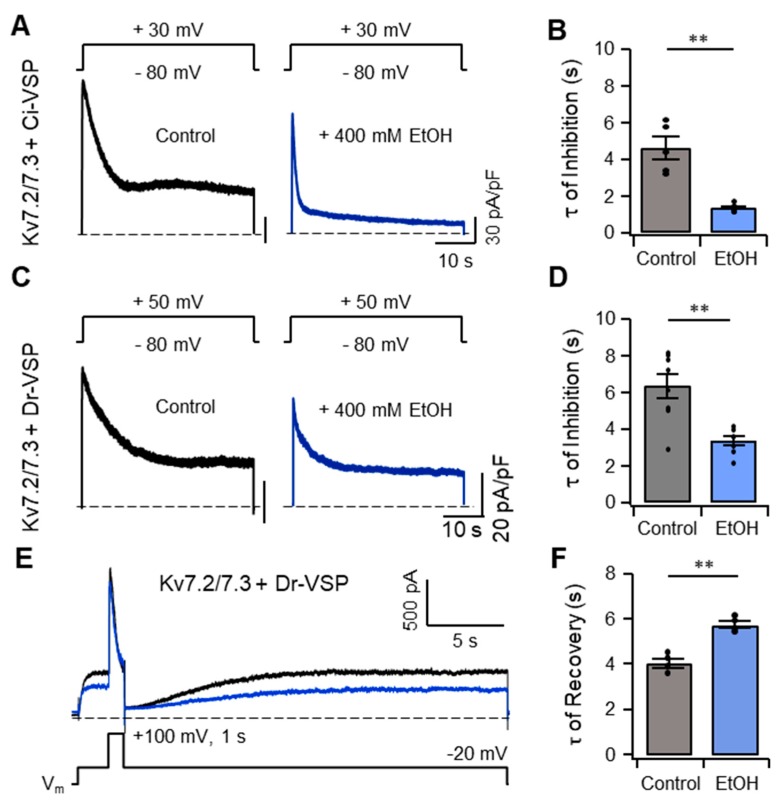
Inhibition of Kv7.2/7.3 currents by ethanol is facilitated by hydrolyzing plasma membrane PI(4,5)P_2_ using a voltage-sensing phosphatase. (**A,C**) Representative current traces exhibited by tsA201 cells transfected with both Kv7.2 and Kv7.3, in combination with *Ciona intestinalis* VSP (Ci-VSP) (**A**) and *Danio rerio* VSP (Dr-VSP) (**C**) in response to increasing membrane potentials to +30 mV and +50 mV in the absence (black symbols) and presence of ethanol (blue symbols), respectively. The dashed line indicates zero current. (**B,D**) Time constants (τ) for inhibiting Kv7 current caused by activation of Ci-VSP (**B**, *n* = 5 each) and Dr-VSP (**D**, *n* = 8 each) in the absence (black bars) and presence of ethanol (blue bars). (**E**) Representative current traces exhibited by tsA201 cells transfected with Kv7.2, Kv7.3, and Dr-VSP in response to changes in membrane potential from −80 to +100 mV, followed by −20 mV in the absence (black symbols) and presence of ethanol (blue symbols). (**F**) Time constants (τ) for recovery of Kv7.2/7.3 current after activation of Dr-VSP in the absence (black bar, *n* = 5) and presence of ethanol (blue bar, *n* = 4). Data are mean ± SEM. Mann–Whitney *u*-test, ** *p* < 0.01.

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
