# Peer review of "Ethanol Elevates Excitability of Superior Cervical Ganglion Neurons by Inhibiting Kv7 Channels in a Cell Type-Specific and PI(4,5)P2-Dependent Manner"

_ijms, 2019, doi:10.3390/ijms20184419_

Round 1
Reviewer 1 Report
The paper titled ‘Ethanol elevates excitability of superior cervical ganglion neurons by inhibiting M/KCNQ channels in a cell type-specific and PI(4,5)P2-dependent manner’ by K-W Kim et al., presents interesting information about modulation of the M-type potassium channel and neuronal excitability mediated by sympathetic system.
I have some questions to be explained:
More detailed information on the statistics function results is needed
Discussion
Page 10, line 1,..physiologically relevant’…, please clarify the expression’
The all abbreviations should be explained, if they occur for the first time
Reviewer 2 Report
The authors present a manuscript on the effect of ethanol on Kv7 channels. The results show an increased excitability of SCG neurons in response to high concentrations of ethanol. The authors claim that an inhibition of currents through Kv7 channels is responsible for this increase. Furthermore, the authors suggest that the ethanol-mediated inhibition of Kv7 currents is influenced by PIP2.
First, I would like to suggest to the authors that they use the most recent uniform nomenclature for proteins introduced by the IUPHAR (https://www.guidetopharmacology.org/GRAC/FamilyDisplayForward?familyId=81) for KV7 channels in favor of M-channel/ M-current/ KCNQ-current/ KCNQ-channel/ KCNQ1-5 subunit etc.
In row 29 to 30, page 4, and row 15, page 10, the authors write about "physiologically relevant concentrations of ethanol". First, there is no "physiological relevance" it should rather be "pharmacological relevance". And the authors might want to explain to the readers what ethanol concentrations are typically reached after alcohol consumption.
It is not clear to me why the authors test a maximum concentration of 100mM ethanol (Fig 3) in SCG neurons but for all recordings in transfected cells 100mM is the minimum concentrataion.
In Fig 4: It is not clearly visible what the maximum ethanol concentration is in Fig 4F, it would be interesting, if the authors could state that concentration. In that respect, it would also be interesting what a current and a time course under these conditions (I assume it is 3 M ethanol) looks like.
Is the activation curve in Fig4G a representative activation curve or the sum of several recordings? If it is several recordings, the authors could state the n-number. If it is a representative activation curve: why did the authors choose to show one representative cell, while in Fig 5A the sum of several recordings is shown?
In Fig 5, the authors show that the inhibition mediated by 30 mM TEA is also influenced by the presence of PIPKgamma. Considering that TEA is a pore blocker (Hadley et al. 2000, PMID: 10711337), can the authors explain how that might occur? Clearly, TEA block should not be influenced by PIP2 availability/ binding.
In Fig 6 and the according results section (page 8), the authors conclude that ethanol inhibition of Kv7 currents inversely correlates with the reported PIP2 affinities of the different subunits (Kv7.2 homomers low PIP2-affinity; Kv7.3 homomers high PIP2-affinity; Kv7.2/3 heteromers intermediate affinity). However (likely by coincidence), it also correlates with the reported TEA sensitivity of the respective subunits (Kv7.2 homomers - high sensitivity towards TEA; Kv7.3 homomers low sensitivity towards TEA; heteromers intermediate sensitivity - Hadley et al. 2000, PMID: 10711337; Shapiro et al. 2000, PMID: 10684873). Both correlations might just be coincidences and if the authors want to show a clearer (inverse) correlation, they might want to investigate Kv7.4 homomeric channels, since they display a similar affinity towards PIP2 as Kv7.2 channels (Li et al. 2005, PMID: 16251430) but a lower TEA sensitivity (Hadley et al. 2000, PMID: 10711337).
In Fig 7, I do not think the tau-recovery is a good readout for PIP2 affinity, since it is not just dependent on re-association of PIP2 with the channels, but also on the reformation of PIP2 mediated by enzymes like PI5K.
Furthermore, I am not sure why the authors use a two-way ANOVA. The authors only compare control current vs. treated current within the same cell - therefore they should perform a paired t-test. And since with an N of 5 a Gaussian distribution can not be reliably tested, a non-parametric test should be performed.
I also have some questions regarding the methods:
Can the authors explain why they chose to perform their experiments in whole-cell mode? It has been known for decades that Kv7 currents exhibit a run-down phenomenon in the conventional whole-cell mode (Brown et al. 1989, PMID: 2689633, Shapiro et al 2000, PMID: 10684873). In order to prevent this phenomenon, people are using the perforated whole-cell mode achieved by amphothericin B (Choveau et al. 2018, PMID: 30348901) or Nystatin (Koyama et al. 2006, PMID: 16956995). The run-down occurs most likely due to loss of PIP2 (Li et al 2005, PMID: 16251430) and, as the senior author has published himself, these ion cannels are highly dependent on the availability of PIP2. So why would one choose to investigate these channels in the conventional whole cell mode, and in particular if the investigated effects are dependent on PIP2 levels?
Maybe the authors would see more pronounced effects by ethanol if they were performing their experiments in the perforated mode?
In addition, I am wondering about the internal solution: the authors write that the pH of the solution was adjusted to 7.4. Was the NaATP and NaGTP added before or afterwards? If they were added afterwards, they might exceed the buffer capacity of HEPES, rendering the pH of the solution acidic. - which in turn has been shown to decrease the run-down phenomenon (Brown et al. 1989, PMID: 2689633) due to unknown reasons.
In general, the methods section could be improved. The description of how activation curves were performed is poor. And it is entirely unclear at which voltage currents were determined.
For example: on page 13, row 10, the methods section reads as: "the current was measured at the holding potential (-80mV)." What is "the current"? If the authors talk about m-currents - there should not be any current - unless it is a tail current - which the authors should mention. I assume this part deals with the recording of activation curves? However, this detail is not mentioned.
Overall, I think the authors investigate an interesting phenomenon, which does, however, not occur at pharmacologically relevant concentrations. This fact should be mentioned by the authors in the discussion.
The authors speculate about amino acid residues that might be involved. Why did the authors not attempt to tackle this question?
Round 2
Reviewer 2 Report
The authors present a revised version of their manuscript entitled "Ethanol Elevates Excitability of Superior Cervical Ganglion Neurons by Inhibiting Kv7 Channels in a Cell Type-specific and PI(4,5)P2-Dependent Manner".
The authors have addressed all my points. I have just one minor comment: In Fig. 6, the authors still write "KCNQ2 and KCNQ3" in their figure - which is now confusing since they renamed all KCNQ and M-channels to Kv7 channels.
I am looking forward to the subsequent publication.
Author Response
We are grateful to reviewer for constructive comment for improvements. We revised the words and changed the figure.
Point 1: The authors have addressed all my points. I have just one minor comment: In Fig. 6, the authors still write "KCNQ2 and KCNQ3" in their figure - which is now confusing since they renamed all KCNQ and M-channels to Kv7 channels.
Response 1: I thank you for your thorough review! Following the suggestion, we changed the channel names to a uniform nomenclature Kv7 in Figure 6. We have also revised the figure legend.